# Emerging View on the Molecular Functions of Sec62 and Sec63 in Protein Translocation

**DOI:** 10.3390/ijms222312757

**Published:** 2021-11-25

**Authors:** Sung-jun Jung, Hyun Kim

**Affiliations:** School of Biological Sciences and Institute of Microbiology, Seoul National University, Seoul 08826, Korea; lifecard@snu.ac.kr

**Keywords:** Sec61, Sec62, Sec63, protein translocation, endoplasmic reticulum

## Abstract

Most secreted and membrane proteins are targeted to and translocated across the endoplasmic reticulum (ER) membrane through the Sec61 protein-conducting channel. Evolutionarily conserved Sec62 and Sec63 associate with the Sec61 channel, forming the Sec complex and mediating translocation of a subset of proteins. For the last three decades, it has been thought that ER protein targeting and translocation occur via two distinct pathways: signal recognition particle (SRP)-dependent co-translational or SRP-independent, Sec62/Sec63 dependent post-translational translocation pathway. However, recent studies have suggested that ER protein targeting and translocation through the Sec translocon are more intricate than previously thought. This review summarizes the current understanding of the molecular functions of Sec62/Sec63 in ER protein translocation.

## 1. Introduction

Approximately one-third of the eukaryotic proteome is directed to the ER for localization in the organelles of the secretory pathway or secreted out of the cells. Most translocated proteins have a signal sequence (SS) or a transmembrane domain (TMD) in their N-terminus, which targets them to the ER. These proteins are thought to be translocated across or inserted into the ER membrane via the Sec61 protein-conducting channel co-translationally or through the Sec complex post-translationally that additionally contains Sec62/Sec63 [1].

However, recent ER-proximity ribosome profiling studies have shown that most ER protein targeting occurs in a co-translational manner [2,3]. In contrast to the ribosome profiling results, cryo-electron microscopy (EM) structures of the yeast Sec complex show that the Sec complex is not compatible for co-translational translocation because the ribosome and Sec63 share the same binding site on the Sec61 channel [4,5,6,7]. In the subsequent sections, we summarize the classical and emerging views on protein targeting and translocation via the Sec complex, and discuss the underlying mechanisms and molecular functions of the Sec62/Sec63 complex in translocation of nascent chains.

## 2. Classical View on Protein Targeting and Translocation in the ER

The signal recognition particle (SRP) recognizes the N-terminal SS or TMD of secretory and membrane proteins when it emerges from the ribosome, arrests translation, and escorts the ribosome-nascent chain-SRP complex to the SRP receptor (SR) in the ER membrane. After delivering the complex to the ER membrane, SRP is released from the ribosome-nascent chain complex, which is then docked on the Sec61 channel. Synthesis of a nascent chain continues from the ribosome; the N-terminal SS or TMD binds to TM helices 2 and 7 of Sec61, the lateral gate, which triggers the opening of the channel and translocation of a nascent chain occurs across the ER membrane. This co-translational translocation pathway has been synonymously referred to as the SRP-dependent targeting and translocation pathway (Figure 1A) [1].

In the post-translational translocation pathway, nascent chains with less hydrophobic SSs are not recognized by SRP and completely synthesized in the cytosol (Figure 1A). Cytosolic Hsp70 chaperones bind to these proteins, preventing premature aggregation of the fully translated proteins and escort them to the Sec complex containing Sec62/Sec63 in the ER membrane. The yeast Sec complex contains additional Sec71(66) and Sec72 subunits, which are not found in mammals [8,9]. Sec63 interacts with an ER lumenal chaperone, Kar2 (Bip in mammals), which facilitates the unidirectional movement of a nascent chain into the lumen [10,11]. Since SRP is not involved here, this pathway has been referred to as the SRP-independent targeting and translocation pathway [1].

## 3. Emerging View on Protein Targeting and Translocation in the ER

A deep-sequence-based proximity-specific ribosome profiling technique has made it possible to investigate the local translation of the proteome. An ER-enriched ribosome profiling study revealed that not only SRP-dependent but also SRP-independent proteins are translated at the ER membrane [3]. Secretome mRNAs are associated with SRP even before the target signal is exposed and targeted to the ER membrane [2]. Thus, these studies suggest that the majority of the secretome is co-translationally targeted to the ER. Subsequent rounds of translation can occur on the mRNAs that are already present after pioneering targeting of the ribosome-nascent chain complex to the ER membrane (Figure 1B). These studies reveal that the SRP-dependent and independent targeting are not synonyms to the co- and post-translational targeting, respectively, implicating that the ER protein targeting is more dynamic and intricate than previously thought.

Hsp70 cytosolic chaperones guide the post-translational translocation of proteins. Among them, Ssb1 (Stress-Seventy subfamily B 1) in yeast is associated with the ribosome-nascent chain complex, mediating the co-translational folding of proteins [12,13,14,15]. Ribosome profiling studies on Ssb substrates revealed that ~40% of all ER-targeted proteins interact with Ssb [16]. Ssb1 also interacts with Sec72 in the Sec complex [17]. These observations suggest an interplay between co-translational targeting and translocation via the Sec complex (Figure 1B).

## 4. The Sec62/Sec63 Complex in Yeast

### 4.1. Discovery

Sec62 and Sec63 in the baker’s yeast, *Saccharomyces cerevisiae*, were identified by genetic screening to isolate yeast mutants that accumulate reporter secretory proteins in the cytosol [18,19]. Translocation of a subset of secretory precursors was defective in Sec62 and Sec63 mutant strains [18,19]. Association of Sec62 and Sec63 with Sec61 channel was determined by chemical cross-linking and immunoprecipitation (IP) [20]. When DSP crosslinking of the yeast’s crude membrane fraction solubilized with 1% Triton-X 100, it was followed by immunoprecipitation with Sec62 antibodies, Sec61 and Sec63 in addition to a glycosylated 31.5 kDa and an unglycosylated 23 kDa proteins were found to be associated. However, Sec61 was no longer associated with Sec62 when the cross-linking reaction was omitted prior to IP. It suggests that Sec62 interacts with Sec61 rather weakly whereas it forms a relatively stable tetramer complex with Sec63, 31.5, and 23 kDa proteins. The 31.5 and 23 kDa proteins were later identified as Sec71(66) and Sec72 in a genetic screening [21].

For reconstitution of the translocation process in vitro, the Sec translocon was purified by solubilization of yeast microsomes with 3% digitonin and subjected to incubation with Concanavalin A beads that specifically bind to glycoproteins, such as Sec71 [22]. This study identified a heptameric Sec complex, consisting of the Sec61/Sbh1/Sss1 trimer and Sec62/Sec63/Sec71/Sec72 tetramer. To determine whether the Sec61 and the Sec62/Sec63 complexes have independent binding activity with a substrate, an in vitro binding assay was carried out by reconstituting the Sec61 and Sec62/Sec63 complexes together or separately with prepro alpha factor (ppαF, yeast mating factor α), a Sec62/Sec63 dependent substrate [23]. Each complex showed weak binding whereas together they showed efficient binding to ppαF as in intact Sec complex, suggesting that the Sec heptamer is a functional complex required for binding of Sec62/Sec63 substrates. Relative stability of the sub-complexes of the Sec translocon was assessed by co-immunoprecipitation of Sec62-HA using the 3% digitonin solubilized microsomes, incubation of the purified Sec complex in buffer containing increasing concentration of Triton-X 100, and separation of the complexes on BN-PAGE [8]. The heptameric complex started to dissociate to the Sec61 trimer and the Sec62/Sec63 tetramer from the Triton-X 100 concentration above 0.4%. The Sec63 trimer containing Sec63/Sec71/Sec72 also appeared from the Triton-X 100 concentration 0.4~0.8%, indicating that Sec62 dissociated from the Sec62/Sec63 tetramer. The Sec61 trimer, the Sec62/Sec63/Sec71/Sec72 tetramer, and the Sec63/Sec71/Sec72 trimer were found in the presence of 1% Triton-X 100. A relatively week interaction of Sec62 with the remaining Sec translocon was also observed in the BN-PAGE analysis of the 2% digitonin solubilized yeast microsomes [24]. Here, the Sec heptamer (SEC complex) and the hexamer lacking Sec62 (SEC’ complex) were resolved on BN-PAGE. These experiments indicate dynamic nature of Sec62 and the possibility of the existence of the Sec complex without Sec62 in the ER membrane.

Meanwhile, Ssh1 (Sec sixty-one homolog 1) was discovered from the sequence homology search. It shares ~50% sequence identity with Sec61 [25]. Ssh1 forms a trimer complex with Sbh2 (Sbh1 homolog) and Sss1 as similar to the Sec61 trimer, but is not associated with the Sec62/Sec63 complex [26]. Hence, the Ssh1 complex has been proposed to function exclusively on the co-translational translocation.

These studies have identified that the Sec62/Sec63 complex function on protein translocation in the ER membrane and characterized the Sec heptameric complex consisting of Sec61 trimer and the Sec63 trimer complexes in addition to a loosely associated Sec62.

### 4.2. Structure

Cryo-EM structures of the Sec complex have been obtained recently [4,5,6,7]. The structures revealed that Sec63 tightly interacts with Sec61 through the cytosolic, membrane, and lumenal domains (Figure 2A–C). On the membrane side, three TMDs of Sec63 were found at the back-side of Sec61, opposite to the lateral gate (Figure 2C). Sec63-Sec61 interaction causes the lateral gate helices to separate; the pore is wider than that observed in any other Sec61/Y structures [27,28,29,30,31,32]. At the cytosolic side, soluble domains of Sec63, Sec71, and Sec72 interact with each other and are located above the Sec61 pore. Sec62 was poorly resolved in the first two structures [6,7]; however, two recent structures mapped Sec62 TMDs at the lateral gate of Sec61 [4,5]. An SS-bound Sec complex shows that the SS is sandwiched between the lateral gate helix 7 of Sec61 and TMD2 of Sec62 on the membrane side [4]. Sec62-bound Sec translocon shows further opening of the pore and displacement of the luminal plug domain in the lumen (Figure 2C).

These structures suggest two important functions of the Sec62/Sec63 complex: (1) its binding to the Sec61 complex opens the protein-conducting channel, and (2) Sec62 is likely the subunit that recognizes the SS and TMD of nascent chains at the lateral gate of Sec61.

### 4.3. Substrate Specificity

A study by Ng et al. showed that a few secretory precursors with low hydrophobic SSs that are not recognized by SRP require Sec62 and Sec63 [33]. Since then, various mutants of the Sec62/Sec63 complex have been characterized, wherein the translocation of certain secretory and membrane precursors is impaired [24,34,35,36]. These studies have helped to deduce the substrate specificities of the Sec62/Sec63 complex. The findings are summarized in the subsequent sections.

#### 4.3.1. Sec62

Sec62 has two TMDs with N- and C-termini facing the cytosol (Figure 2A) [18]. The N-terminus of Sec62 contains a cluster of basic amino acids. When these residues were substituted with acidic residues (Sec62_35DDD), the interaction of Sec62 with Sec63 was disrupted. The cytosolic, C-terminal flanking region of the second TMD has been proposed to constitute a potential SS binding site [36,37]. When three residues in this region were replaced with alanines, yeast cells bearing this mutation no longer survived, and those bearing single residue replacement to alanine showed a growth defect (Sec62_P219A) [34]. However, the interaction between Sec62_P219A mutant and Sec63 was intact, unlike the Sec62_35DDD mutant [38]. Recent cryo-EM structure shows that this region is important in anchoring the Sec62 TMD2 in the cytosolic side of the ER membrane [5]. In these Sec62-mutant strains, translocation of secretory precursors and membrane proteins with moderately hydrophobic TMDs was defective [34].

For signal-anchored proteins, two modes of topogenesis have been shown: (1) the TMD inserts as the N-terminus facing the lumen (head-on) and then reorients to form the final topology, and (2) the TMD inserts as the N-terminus facing the cytosol, forming a loop conformation with the downstream region [39,40]. Our recent study showed that topogenesis of moderately hydrophobic N-terminal signal-anchored proteins was defective in the Sec62_35DDD mutant [41]. While the head-on insertion of the test N-terminal signal-anchored protein occurred, the subsequent reorientation step was defective in the Sec62_35DDD mutant but not in the Sec62_P219A mutant, indicating that association of Sec62 with Sec63 is needed for this step [41]. When positively charged residues were introduced in the C-terminal flanking region of the signal anchor of the test protein, making its inversion slower, topogenesis of the test protein became defective in both Sec62 mutant strains, suggesting that Sec62 is especially needed in inversion of the signal anchor with unfavorable topogenic signal [41]. The head-on inserted form of the test signal-anchored protein was co-immunoprecipitated with the Sec62 mutant; therefore, we proposed a model in which Sec62 recognizes the head-on inserted signal-anchored protein and mediates its reorientation as N_in_-C_out_ membrane topology.

#### 4.3.2. Sec63

Sec63 has three TMDs with the N- and C-termini facing the ER lumen and cytosol, respectively (Figure 2A). The lumenal loop between the second and third TMDs contains the DnaJ domain, through which an ER lumenal chaperone, Kar2 (Bip in mammals) interacts with. The DnaJ-domain of Sec63 is indispensable for the translocation of both SRP-dependent and -independent precursors (a test membrane protein with a hydrophobic TMD and secretory proteins having moderately hydrophobic SSs) [11,42,43].

Truncation of the N-terminal 40 residues including the first TMD of Sec63 destabilizes the Sec complex, judging by BN-PAGE and impaired insertion of membrane proteins [38]. Sec63 has a large cytosolic C-terminal region that associates with Sec62, Sec71 and Sec72. Deletion of the FN3 (or the Brl) domain in the cytosolic region impairs the assembly the Sec complex and translocation of both SRP-dependent and -independent substrates (test precursors with hydrophobic TMD and those with less hydrophobic cleavable SS, respectively) [6,7,24]. The C-terminal end is enriched with acidic amino acids that interact with the N-terminal basic residues of Sec62 [24,44]. Threonine at positions 652 and 654 at the C-terminus can be phosphorylated, strengthening the interaction with Sec62 [45]. Deletion of the acidic region of Sec63 impairs the translocation of Sec62-dependent substrates. Sec62 is required for the translocation of precursors with moderately hydrophobic SS or TMDs, whereas Sec63 is required for translocation and membrane insertion of most test proteins regardless of their SS hydrophobicity, implicating its general role in translocation of all types of proteins in yeast.

#### 4.3.3. Sec71 and Sec72

Sec71 is a single-pass membrane protein with N_out_-C_in_ orientation, and its C-terminus interacts with Sec72 (Figure 2A) [4,5,6,7,16]. Sec72 has a tetratricopeptide repeat (TPR) domain that binds to cytosolic Hsp70 chaperones, Ssa1 (Stress-Seventy subfamily A 1) and Ssb1 [15,17]. While Ssa1 binds to fully translated proteins, Ssb1 associates with translating ribosomes [13]. Mutations in the TPR domain of Sec72 lead to defects in its interaction with Ssa1 and Ssb1 and cause a translocation defect in vacuolar carboxypeptidase Y (CPY) [17].

In the systematic assessment of the Sec62/Sec63 dependent SS characteristics, the SS of CPY, a representative secretory protein, varied in its hydrophobicity and the length of the N-terminus preceding the SS hydrophobic core. Translocation efficiencies of CPY variants at an early stage were assessed using 5-min metabolic labeling of Sec62-, Sec63-defective and Sec71-, Sec72-deletion cells [46]. Deletion of Sec72 affected the translocation of a subset of CPY variants with less hydrophobic SSs, as observed in the Sec62 mutant strain. In comparison, translocation of the CPY variants with hydrophobic internal SSs, which are not dependent on Sec62, was severely impaired in the Sec71 deletion strain. A ribosome profiling study showed that targeting and translocation of precursors with internal SSs were defective in the Sec71(66) deletion strain [3]. These data suggest that Sec71(66) is involved in mediating translocation of precursors having internal SSs that insert as a loop conformation (Figure 3).

Observations that substrate specificities differ among the four components of the Sec62/Sec63 complex suggest the possibility that individual components of the Sec62/Sec63 complex may have distinct functions in aiding translocation, membrane insertion, and/or folding of different types of incoming nascent chains.

## 5. The Sec62/Sec63 Complex in Higher Eukaryotes

### 5.1. Discovery

Sec62 of *Drosophila melanogaster* was discovered as Dtrp1 (Drosophila translocation protein 1) [47]. Dtrp1 rescues defects in cell growth and protein translocation due to Sec62 deletion in yeast. Thereafter, human Sec62 was identified by sequence homology to *Drosophila* Sec62 (HTP, human translocation protein 1) [48]. Human Sec63 was identified by sequence homology of the human cDNA to the yeast homolog [49].

Two groups have reported that Sec62 and Sec63 are associated with the Sec61 complex in bovine and dog pancreas rough microsomes [50,51]. Meyer et al. showed that Sec62 and Sec63 are ubiquitously expressed in all rat and bovine tissues. The C-terminal acidic residues of Sec63 were found to interact with the N-terminal basic residues of Sec62 in the cytosolic side of the ER membrane as in yeast, and expression of human Sec62 rescued the growth defect of yeast cells carrying a defective Sec62, demonstrating that human and yeast Sec62s are structural and functional homologs [52].

### 5.2. Substrate Specificity

#### 5.2.1. Small Proteins

Translocation of secretory precursors shorter than ~160 amino acids was impaired in mammalian cells depleted of Sec62 [53]. Interestingly, longer proteins were defective in translocation in the cells depleted of SRP receptor α (SRα) but not in the Sec62 depleted cells whereas shorter proteins were defective in translocation in the Sec62 depleted cells but not in the SRα depleted cells. Small proteins were partially translocated post-translationally in vitro. This study demonstrated functional conservation of Sec62 in post-translational translocation of secretory proteins in mammals as in yeast. However, unlike in yeast where precursors having less hydrophobic SSs are Sec62 dependent, small size precursors regardless of the characteristics of SSs are found to be dependent on Sec62 in mammals.

When different test proteins of varying hydrophobicity and the C-terminal length were assessed for their translocation efficiency in microsomes isolated from or semi-permeabilized human cells depleted of Sec62, post-translational translocation of preproapelin, a small secretory protein, was reduced, whereas both co- and post-translational translocation of preproapelin was impaired in the cells depleted of Sec63 [54,55]. The dependence of Sec62 and Sec63 was lost when the C-terminus of preproapelin was lengthened with dihydrofolate reductase (DHFR).

#### 5.2.2. Signal Sequence Characteristics

Potential Sec62 and Sec63 substrates were searched using quantitative mass spectrometry analysis of proteomes from Sec62 or Sec63 knocked-down and knocked-out human cells [56,57]. Although not many were found, negatively affected proteins contain less hydrophobic SSs [56]. Post-translational translocation of preproapelin, which has relatively less hydrophobic SSs, was impaired in human cells depleted of Sec62 and Sec63 [54].

#### 5.2.3. Mature Domain Region

For preproapelin, positively charged residues downstream of the SS are important for Sec62- or Sec63-dependent translocation [54]. When these residues were substituted to eliminate positive charges, translocation efficiency improved in the Sec62/Sec63-depleted cells. ERdj3, another Sec62 and Sec63 substrate, and prion protein have positively charged residues in their mature domains adjacent to the SS that affect their dependency on Sec62/Sec63 and Sec63, respectively [56,58]. In the Sec62 or Sec63 depleted cells, a pre-ERdj3 form was found in the membrane pellets upon carbonate extraction, suggesting that the head-on inserted precursor was unable to reorient in the absence of Sec62/Sec63.

#### 5.2.4. Secretory Precursors That Are Inhibited by CAM741 in the ER Translocation

The cyclic heptadepsipeptide CAM741 (CPD A) is a selective translocation inhibitor for a subset of secretory precursors [55,59]. Secretory precursors with lower hydrophobicity and positively charged residues downstream of the SS, the types that depend on Sec62/Sec63, were especially sensitive to CAM741.

## 6. Additional Functions of Sec62 and Sec63 in Mammals

### 6.1. Association of Sec62 with Ribosome

Human Sec62 has longer N- and C-termini than the yeast homolog. Residues 1–15 and 156–170 of human Sec62 contain clusters of positively charged residues that resemble the ribosome binding domain found in other ribosome-interacting proteins [52]. An in vitro-binding study showed that the N-terminal cytosolic fragment of Sec62 binds to ribosomes [52]. Yeast Sec62 does not bind to the ribosome, but when the N-terminal 12 residues of human Sec62 were fused to the yeast homolog, it bound to the ribosome.

However, Sec62/Sec63 was co-immunoprecipitated with Sec61β in the ribosome-free fraction of bovine microsomes [51]. Muller et al. suggested that the failed interaction of Sec62 with ribosomes in the earlier study [51] may be due to the high salt concentration (400 nM) in the microsome solubilization buffer as the binding of the N-terminal fragment of human Sec62 and the ribosome weakened when the salt concentration was higher than 300 nM [52].

To capture the interacting partners of the translating ribosome-nascent chain, Conti et al. designed an experiment using a translation-arrested nascent chain at increasing length on the ribosome and detected using BN-PAGE [60]. The ribosome-bound prion precursor, which has an inefficient SS, was found to be associated with Sec62/Sec63 in addition to Sec61.

### 6.2. Competitive Binding of Sec62 and SR to Sec61

Chemical crosslinking of rough microsomes and co-immunoprecipitation with Sec61β pulled down the SRα subunit, Sec61α, Sec61β, and SPC25, a subunit of the signal peptidase complex [61]. When purified SRα was added to the microsomes, crosslinking between Sec61β and Sec62 was reduced, whereas crosslinking between Sec61β and SRα was enhanced, suggesting competitive binding between Sec62 and SRα to Sec61β. The Sec61 structures show that Sec61β is positioned near the lateral exit site where the TM segment is inserted into the membrane, indicating that multiple components (SRα, Sec62, SPC25) dynamically associate with the Sec61 complex near the lateral exit via Sec61β. The functional relationship between Sec62 and SR has also been observed in the quantitative mass spectrometry analysis of Sec62-depleted human cells. The abundance of SR subunits was upregulated upon depletion of Sec62 [56].

### 6.3. Role of Sec62 in Autophagy

Autophagy receptors are characterized by an LC3-interacting region (LIR). Binding of the ubiquitin-like protein LC3-II to LIR triggers selective autophagy [62]. The conserved LIR motif was found in the C-terminus of human Sec62 (NDFEMIT, residues 461–367) and is important for Sec62-mediated selective autophagy of the ER components after an unfolded protein response. This region is dispensable for protein translocation function and is not present in yeast Sec62.

### 6.4. Sec63 Interacting Proteins

In addition to Sec61 and Sec62, co-immunoprecipitation of Sec63 using the detergent solubilized dog pancreas microsomes yielded two additional proteins, calumenin and reticulocalbin [50]. They are calcium binding proteins residing in the ER lumen and have EF hand motifs [63,64]. A number of studies has shown that calcium leaks from the ER lumen to the cytosol via the Sec61 translocon [65,66,67,68,69], hinting the interplay between calcium and the translocon. However, the functional significance of the Sec63 interaction with these calcium binding proteins awaits to be revealed.

Further, a yeast two-hybrid screening of a human placenta cDNA library using the C-terminal domain of human Sec63 as a bait identified a cytosolic nucleoredoxin (NRX) [70]. GST pull-down and peptide-binding assays between the C-terminal region of Sec63 (residues 509 to 559 within the Brl domain) and the C-terminal part of NRX (residues 411 to 430) confirmed their interactions. Since NRX is involved in the Wnt signaling pathway, the authors suggested a possible link between Sec63 and Wnt signaling.

## 7. Human Diseases Associated with Sec62/Sec63

Considering that the Sec61 channel and its associated protein complexes mediate early protein biogenesis of about 30% of the proteome, it is not surprising that defects in the components of these machineries are linked to various diseases [71,72,73].

Elevated expression of Sec62 has been observed in some cancer tissues [74,75,76,77,78,79,80]. Hence, Sec62 has been suggested as a potential cancer marker, cancer-causing, or anti-cancer drug resistant factor, although whether the expression levels of Sec62 influence protein translocation and how the changes of its abundance are related to cell physiology and the development of diseases remain elusive. Since human Sec62 has dual functions as a translocation component and a receptor for ER-specific autophagy, its role in cancer requires further investigation.

Sec63 is found to be linked to diabetes, cancers, and autosomal dominant polycystic liver and kidney diseases [71,72,73]. Autosomal dominant polycystic liver and kidney diseases result from defective biogenesis of polycystin 1/2 that are cilia membrane proteins and function as a calcium-permeable receptor-channel complex. Sec63 is involved in the biogenesis of polycystin 1/2, and cyst formation in the liver is increased in Sec63 defective mice [81,82,83]. Sec63 interacts with calcium binding proteins in the lumen and nucleoredoxin in the cytosol, thus it is possible that signaling pathways through these interacting proteins (impaired interactions with defective Sec63) may contribute to the development of diseases. Human diseases associated with Sec translocon are summarized in [71,72,73].

## 8. Conclusions

Studies to date have shown that evolutionarily conserved Sec62 mediates the translocation of proteins with specific characteristics. These characteristics are SSs and TMDs with low hydrophobicity and poor topogenic signals in yeast, and small size, moderately hydrophobic SSs, or the presence of positively charged residues in the downstream of the SS in mammals (Figure 3). Precursors having these characteristics insert as head-on (N_out_-C_in_ orientation), and Sec62/Sec63 mediate inversion of the head-on inserted SS and TMD as in N_in_-C_out_ orientation. The cryo-EM structures of the yeast Sec complex show that Sec62 is located at the lateral gate of Sec61 where the SS and TMD bind to, suggesting that it recognizes the SS or TMD features of Sec62 clients [4,5]. Sec63 binds at the back-side and the cytosolic side of the Sec61 translocon, widening the pore of the Sec61 channel (Figure 2C) [4,5,6,7]. These studies collectively suggest that Sec62 and Sec63 function in Sec61-channel gating for those proteins that are insufficient to open the channel on their own.

## 9. Perspectives

In spite of the progress in elucidating the functions of Sec62/Sec63 in protein translocation, unresolved questions remain to be addressed in future studies. Biochemical and structural studies indicate that Sec62 is flexible, thus interacts with the Sec complex less tightly compared to the other subunits. Sec63 mediates translocation of broader types of precursors than Sec62. Although Sec62/Sec63 forms a complex, these observations raise the question of whether the Sec complex lacking Sec62 exits and functions in the translocation of Sec62-independent precursors. Further, it is unclear whether certain features of the Sec62/Sec63 clients are recognized by distinct components/domains of the Sec62/Sec63 complex or whether these features cause the nascent chain to be in particular intermediate forms (e.g., head-on inserted form), which are then recognized by Sec62/Sec63.

The CAM741 selectively inhibits translocation of a subset of secretory precursors possessing similar features as the Sec62/Sec63 clients [54]. It is elusive how the CAM741 achieves such substrate selectivity for the general translocon that handles a bulk of proteins that pass through the ER membrane. It raises the question of whether CAM741 inhibits proper association of Sec62/Sec63 with Sec61 and the incoming nascent chain, thus selectively impairing ER translocation of Sec62/Sec63 clients.

Sec71 and Sec72 were first discovered from a genetic screen searching the components involved in membrane protein biogenesis [21]; however, their functions in membrane protein biogenesis as well as the existence of their functional homologs in higher eukaryotes remain to be revealed.

Studies have shown that human Sec62 can bind to ribosomes and yeast Sec72 interacts with the ribosome-nascent chain associated Ssb1 [17,52], implying the function of the Sec complex in co-translational translocation. The underlying mechanisms of how the Sec complex mediates co-translational translocation in the ER membrane await further investigation.

Lastly, expression levels of Sec62 and Sec63 are found to be altered in various cancer cells [74,75,76,77,78,79,80]. Future studies are needed to clarify how the expression levels of Sec62 and Sec63 influence protein translocation, and how the changes of their abundance are related to cell physiology and the development of diseases.

## Figures and Tables

**Figure 1 ijms-22-12757-f001:**
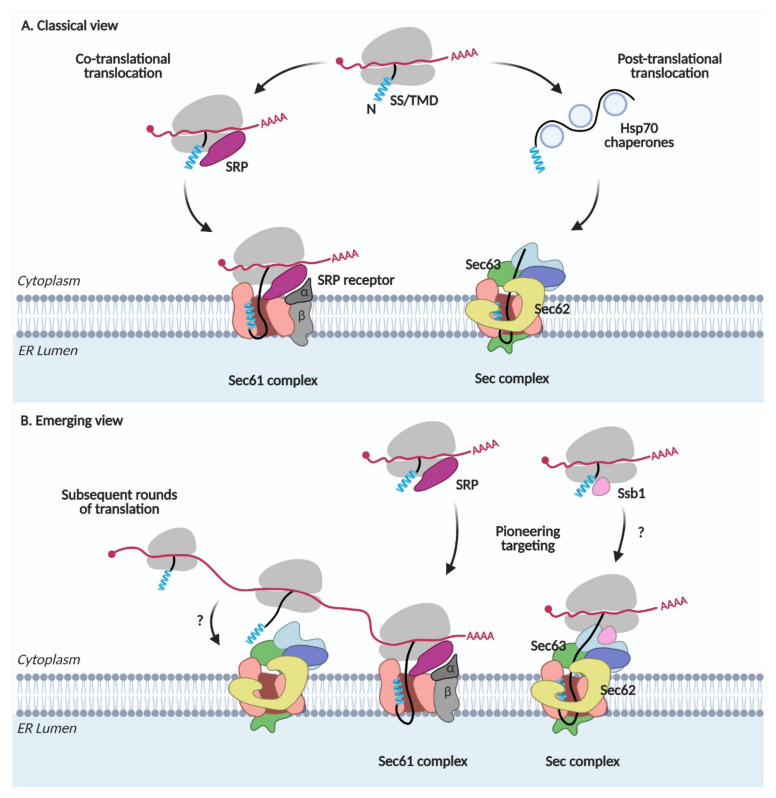
Protein targeting and translocation in the ER. (**A**) Classical view. SRP, signal recognition particle; SS, signal sequence; TMD, transmembrane domain. (**B**) Emerging view.

**Figure 2 ijms-22-12757-f002:**
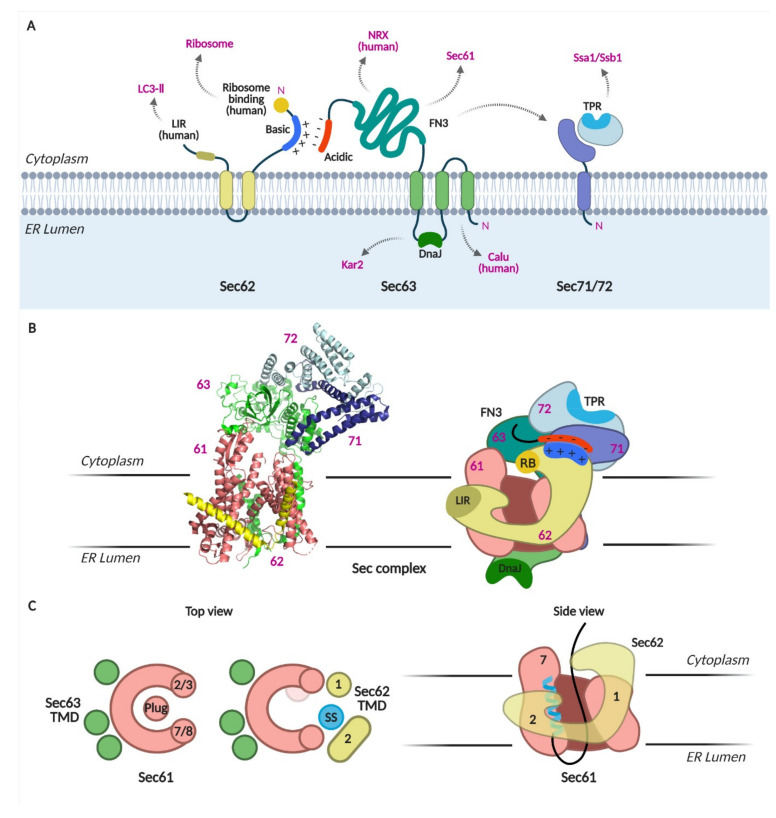
The Sec complex. (**A**) Schematics of Sec62, 63, 71, and 72. Domains and interacting proteins are indicated. LIR, LC3-interacting region; TPR, tetratricopeptide repeat; FN3, fibronectic type III; NRX, nucleoredoxin; calumenin, calu. (**B**) Cryo-EM structure of the Sec complex from *S. cerevisiae*. Itskanov, S., Park, E. (2020) Cryo-EM structure of the Sec complex from *S. cerevisiae*, wild-type, class with Sec62, conformation 1 (C1) doi: 10.2210/pdb7KAI/pdb (PDB code 7KAI) created with PyMOL [5]. Schematic of the Sec complex is based on the cryo-EM structure in [5]. (**C**) Channel opening by binding of Sec63, 62 and signal sequence to Sec61 [4].

**Figure 3 ijms-22-12757-f003:**
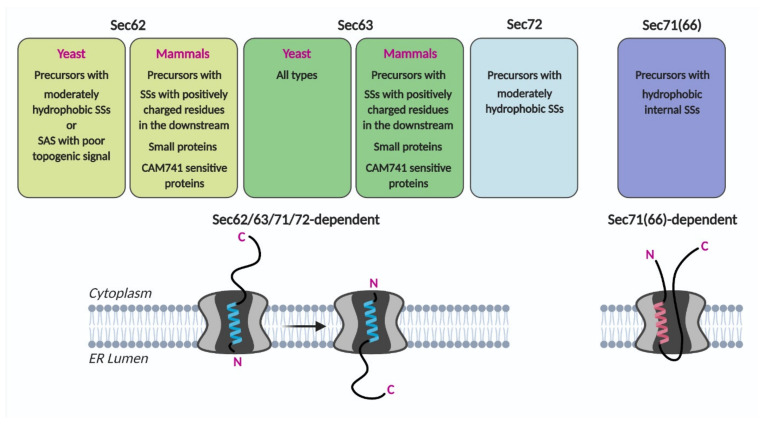
Sec62/Sec63 substrate specificity. Precursors having indicated characteristics may preferably insert the translocon as head-on and then invert (Sec62/63/71/72-dependent) or as a loop conformation (Sec71(66)-dependent). SS, signal sequence; SAS, signal-anchored sequence.

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
