# Peer review of "Emerging View on the Molecular Functions of Sec62 and Sec63 in Protein Translocation"

_ijms, 2021, doi:10.3390/ijms222312757_

Round 1

Reviewer 1 Report

Drs Jung and Kim submitted a comprehensive, well illustrated, and timely update on what is known about the structure and function of the two ER membrane proteins, which are termed Sec62 and Sec63 because of their function in protein secretion. In the first part of their review they focus on the yeast proteins since they were dicovered first, are structurally characterized, and functionally studied best. In the second part, they also report on what is known about the function of the mammalian orthologs and how they are linked to human medicine.

I have only a couple of minor suggestions for improvement:

- In the Abstract the word `membrane´ may be missing after the word `ER´ .

- In my opinion `translation of a nascent chain or of a protein´ is lab slang, since it´s the mRNA which is translated; Therefore, `synthesis´  should be used instead throughout the manuscript.

- Along the same lines: `cytoplasm´ should be replaced by `cytosol´ throughout.

Author Response

I have only a couple of minor suggestions for improvement:

- In the Abstract the word `membrane´ may be missing after the word `ER´.

--> 'membrane' has been added after 'ER'

- In my opinion `translation of a nascent chain or of a protein´ is lab slang, since it´s the mRNA which is translated; Therefore, `synthesis´  should be used instead throughout the manuscript.

--> 'translation' has been changed to 'synthesis'.

- Along the same lines: `cytoplasm´ should be replaced by `cytosol´ throughout.

--> 'cytoplasm' and 'cytoplasmic' have been changed to 'cytosol' and 'cytosolic', respectively, in the revised text.

Reviewer 2 Report

Jung and Kim comprehensively described the molecular function of Sec62 and Sec63 in protein translocation and explore the Sec complex mechanisms. The article is well-written overall. I personally felt that in places the discussion was pretty convoluted and could be simplified for the readers.

Minor comments:

  1. Full form of Ssb1 on page 3 if available.
  2. Correct the reference formatting/spacing.

Author Response

Jung and Kim comprehensively described the molecular function of Sec62 and Sec63 in protein translocation and explore the Sec complex mechanisms. The article is well-written overall. I personally felt that in places the discussion was pretty convoluted and could be simplified for the readers.

--> Discussion (conclusions and perspectives) has been revised.

Minor comments:

  1. Full form of Ssb1 on page 3 if available.

- Full form of Ssb is 'Stress-Seventy subfamily B' and added in the text.

  1. Correct the reference formatting/spacing.

- Reference format has been changed to the MDPI style